# Enhancing the Radar Cross-Range Resolution in Ultra-Fast Radar Scans by Utilizing Frequency Coded Sub-Channels

**DOI:** 10.3390/s22093343

**Published:** 2022-04-27

**Authors:** Christoph Baer, Nicholas Karsch, Robin Kaesbach, Thomas Musch

**Affiliations:** Institute of Electronic Circuits, Ruhr University Bochum, Universitaetsstr. 150, 44801 Bochum, Germany; nicholas.karsch@rub.de (N.K.); robin.kaesbach@rub.de (R.K.); thomas.musch@est.rub.de (T.M.)

**Keywords:** scanning radar, radar resolution, frequency coding, FMCW

## Abstract

This contribution handles a single-channel radar method that utilizes frequency-coded sub-channels for enabling cross-range resolution. Because of the sub-channel coding, the whole area of interest (AOI) is scanned within a single radar measurement. To further enhance the cross-range resolution, the sub-channels’ antenna beams are overlaid in this work, resulting in multiple coding signatures. Next to the operation theory, hardware components, such as coding filters and antennas, as well as signal processing methods, are presented and discussed in detail. A final measurement campaign that investigates several radar scenarios reveals high detection properties and proves the applicability of the proposed radar method.

## 1. Introduction

In the last decades, radar systems found their way into various economic sectors like the process industry, civil air traffic control, security, and civil protection. For each sector, numerous applications utilize outstanding radar properties like robust measurement behavior, through-the-wall measurement applicability, fast and high-resolution measurements as well as scanning applications for the visualization of multi-dimensional scenarios. Here, numerous concepts can be found in the literature that enable different scanning properties that all provide certain advantages but also drawbacks. In the simplest case, a two-dimensional radar scan is performed by mechanically steering the antenna beam. In [1,2,3,4], the authors introduce several mechanical beam steering antennas for industrial, level-probing, and surface reconstruction applications. Because of the time-multiplex operation and slow mechanical reconfiguration, the resulting scanning time duration is on the scale of several seconds. Although this might be sufficient for the intended industrial-level probing applications, the scanning time is too slow for highly dynamic applications. In order to solve the scanning time issues, electronic beam steering antennas, as presented in [5,6,7], are an excellent alternative. Compared to mechanical beam steering, these antennas offer fast beam steering operation in the range of milliseconds. However, as the latter antenna types rely very often on varactor diode-based, phase-modulated antenna arrays [8], integrated, electronic lenses [9], or further reconfigurable transmit arrays [10], electronic beam steering always demands complex and costly setups. In order to combine the benefits of both worlds, the authors of [11,12] propose combined approaches, which utilize so-called electro-mechanical beam steering. To further increase the scanning speed, MIMO-radar and sparsed MIMO systems [13,14,15,16] are the latest development in the state of the art. With almost instantaneous scanning speeds, MIMO radars are among the fastest and most precise radar systems available. Unfortunately, system complexity obviously raises with scanning speed and utilized channels, because every sub-channel requires its own transceiver electronic. Consequently, MIMO-radar systems are too complex and costly for a lot of applications. For enhancing scanning time in signal processing of radar scanning applications, the authors of [17] propose ANN-based beamforming of Microstrip antennas.

The above-mentioned methods have in common that the actual scanning process is performed in time-multiplex, which means that scans of different areas are carried out consecutively. Consequently, the overall scanning time hardly relies on the number of scanning directions and the system’s reconfiguration speed. This however leads to the obvious contradiction of scanning resolution, system complexity, and scanning speed. In order to overcome this drawback, the time-multiplex scanning philosophy can be changed to a frequency- or code-multiplex, leading to instantaneous scanning and single measurement scanning coverage. In [18,19], the authors propose a dispersive scanning antenna, which distributes the available bandwidth to different radiation directions. In terms of scanning speed, this approach is promising, because a single radar measurement covers the whole AOI. Unfortunately, the proposed bandwidth distribution leads to strongly reduced range resolution as the total bandwidth was split up and is reduced in every cross-range direction. Most recently, a radar channel frequency coding method was introduced in [20], which overcomes this drawback. Here, the scanning directions are divided into radar sub-channels, each containing a unique, passive coding filter and antenna. All sub-channels are summed up so that a single measurement from a radar transceiver covers the whole AOI by preserving maximum range resolution. However, in this previous work, antenna beams were strictly decoupled to avoid cross-talk. Consequently, every scanning direction demanded a unique sub-channel. Hence, the cross-range resolution cells corresponded to the number of sub-channels, leading to complex hardware requirements.

Therefore, we propose an advanced, cross-range resolution enhancing method in this contribution. By investigating radar beam overlapping areas, the amount of cross-resolution cells can be increased by keeping the number of sub-channels constant. Moreover, we introduce several hardware components which are mandatory for the proposed method and discuss their performances in detail. Finally, an unsupervised signal-processing method is introduced and compared to the supervised signal processing from [20] in an open range test scenario. The work is structured as follows: Section 1 explains the fundamentals of the radar-channel frequency coding method and discusses possible multi-target scenarios. In Section 2, essential hardware components, i.e., antennas and coding filters are presented, and their performance is demonstrated in simulations and measurements. While Section 3 introduces two different signal processing methods for target classification, Section 4 proves the proposed concept in a wide range measurement campaign. Finally, Section 5 and Section 6 discuss the presented results and conclude the paper, respectively.

## 2. Concept

### 2.1. Fundamentals

The following part explains the basic idea of the frequency coded radar sub-channel concept as it is schematically shown in Figure 1. The initial point is a bi-static radar system with divided transmit (TX) and receive (RX) paths. Although we will utilize a frequency-modulated continuous-wave (FMCW) radar in the later measurements, the fundamental radar system design is not of relevance as long as complex-valued raw data of the transfer function is recorded and available for signal processing. Following the TX path, the transmit signal $s_{t}\left(t\right)$ is fed to a power amplifier (PA) and a sub-subsequent radar transmit antenna. While the TX antenna is chosen in a way that the whole AOI is illuminated, the PA ensures sufficient signal power. In contrast to the TX antenna, which provides a broad beam pattern, the RX antenna array consists of narrow beam antennas that point to different directions within the AOI [20]. Each RX antenna is connected to its own RX filter with a unique filter function within the illustrated filter bank. This approach can also be interpreted as the coding of the radar sub-channel and alignment of the corresponding cross-range direction. Finally, all filter bank outputs are combined into a single RX signal, which is connected to the RX path of the radar system. As combining losses increase with an increasing number of sub-channels, low noise amplifiers as initial stages of the filter bank are useful for multiple sub-channel designs.

The range resolution, i.e., the minimum distance between two equally strong targets that allows for target separation in the radar diagram, $\Delta R_{r}$ of a radar is defined by:(1)$$\Delta R_{r}=\frac{c\cdot k}{2\cdot B}.$$

In (Equation 1), *c* is the speed of light, *B* is the utilized bandwidth, and *k* is a forming factor of the applied window function in signal processing. Obviously, the utilized signal bandwidth has a direct influence on the range resolution. Consequently, the RX-filter design has a significant influence on the later radar performance and resolution. Therefore, in order to preserve maximum range resolution, it is advisable that RX-coding filters should not narrow the absolute signal bandwidth. Hence, the chosen filters could be of the notch or allpass type.

Using the described setup allows for an instantaneous scan of the whole AOI with a single measurement. The receiving signal $s_{r}\left(t\right)$ is the summation of all signals of the *N* sub-channels:(2)$$s_{r}\left(t\right)=\sum_{i=1}^{N}s_{r,i}\left(t\right),$$
while every sub-channel signal $s_{r,i}\left(t\right)$ can be modeled as follows:(3)$$s_{r,i}\left(t\right)=s_{t}\left(t\right)*h_{\mathrm{pa}}\left(t\right)*h_{\mathrm{tx}}\left(t\right)*h_{\mathrm{fs}}\left(t-\tau \right)*\sigma *h_{\mathrm{rx}}\left(t\right)*h_{i}\left(t\right).$$

In (Equation 3), $s_{t}\left(t\right)$ represents the transmit signal while $h_{\mathrm{pa}}\left(t\right)$, $h_{\mathrm{tx}}\left(t\right)$, $h_{\mathrm{fs}}\left(t-\tau \right)$, $h_{\mathrm{rx}}\left(t\right)$, and $h_{i}\left(t\right)$ describe the time domain transfer functions of the PA, the TX antenna, free space propagation, RX antenna, and *i-th* coding filter, respectively. The connecting mathematical operation “∗” is the convolution operator. Furthermore, the parameters σ and τ stand for the target’s radar cross section (RCS) and the time of flight which is defined by
(4)$$\tau =\frac{2R}{c}.$$

Here, *R* is the line-of-sight (LOS) distance to the target. In good approximation, we may consider that the system-related transfer functions do not alter while in operation. Therefore, they will be summarized in the following and described by the system’s time domain transfer function $h_{\mathrm{sys}}\left(t\right)$:(5)$$h_{\mathrm{sys}}\left(t\right)=h_{\mathrm{pa}}\left(t\right)*h_{\mathrm{tx}}\left(t\right)*h_{\mathrm{rx}}\left(t\right)$$

Obviously, the final receiving signal is a 1-dimensional signal, which does not directly provide information on the investigated 2-dimensional scene. Thus, the 2-dimensional signal separation in range and cross-range direction must be performed in signal processing. While a first separation is performed in a range direction, which is equal to a time gating operation, identified targets carry the information of the unique RX-coding filter transfer functions. After obtaining this information, the detected targets are allocated to the corresponding arrival directions. As a result, the AOI is subdivided into a 2D grid, which is scanned instantaneously by a single radar measurement. The corresponding resolution cells depend on the utilized bandwidth in the range direction and the number and beamwidth of the RX antennas in the cross-range direction. For enhancing the cross-range resolution, the RX-antennas are arranged in a way so that the AOI is perfectly subdivided by their patterns while neighboring beams are overlayed by 1/3 of their 3 dB beamwidth. Targets, which are located in the overlapping areas, are carrying the transfer functions of both corresponding RX-coding filters. This results in an increase of cross-range resolution cells to $N_{x}$ resolution cells, depending on a closed or non-closed AOI:(6)$$N_{x}=\left\{\begin{array}{cc} 2N & 360\;\mathrm{degree}\;\mathrm{closed}\;\mathrm{AOI}\\ 2N-1 & \mathrm{non}-\mathrm{closed}\;\mathrm{AOI} \end{array}\right.$$

### 2.2. Single- and Multi-Target Scenarios

Figure 2 shows a schematical drawing of the proposed antenna arrangement for enhancing the cross-range resolution for three antennas and corresponding coding filters in a non-closed setup. While the different beam colors represent the spatial distribution of the antenna arrangement, overlap beams are colored in the corresponding mixed color. The inserted stars represent radar targets in different range resolution cells I to V. This chosen scenario perfectly illustrates all five occurring target interference cases. For the discussion of these cases, we can assume in good approximation that all radar targets provide equal RCSs σ and do not interfere with each other.

#### 2.2.1. Scenario I

Scenario I represents the very basic radar scenario: A single target is located within the main beam of one antenna. In the displayed case, target #1 is located in the main beam of antenna #2. Hence, the corresponding time-domain RX signal can be set up to be:(7)$$s_{r,I}\left(t\right)=s_{t}\left(t\right)*h_{\mathrm{sys}}\left(t\right)*h_{\mathrm{fs}}\left(t-\tau_{I}\right)*\sigma *h_{2}\left(t\right)$$

By means of adequate signal processing, target #1 can be easily found within the radar diagram and the subsequent spectral analysis will deliver the unique filter coding function $H_{2}$ of the corresponding radar sub-channel.

#### 2.2.2. Scenario II

Comparably to the previous description, scenario II also only provides a single target. This time however, target #2 is located in the overlap area of the radar sub-channels #1 and #2. Therefore, the combined RX signal will provide both filter signatures of coding filters #1 and #2, which superimpose to:(8)$$s_{r,\mathrm{II}}\left(t\right)=s_{t}\left(t\right)*h_{\mathrm{sys}}\left(t\right)*h_{\mathrm{fs}}\left(t-\tau_{\mathrm{II}}\right)*\left(h_{1}\left(t\right)+h_{2}\left(t\right)\right)$$

After signal processing, both coding functions are found which leads to the interpretation that the target must be located in the overlap area.

#### 2.2.3. Scenario III

Scenario III is the first scenario that provides two targets in the same range resolution cell, so that a time gating target separation is not possible. Moreover, targets #3 and #4 are located in two neighboring main beam areas, leading to the following RX signal:(9)$$s_{r,\mathrm{III}}\left(t\right)=s_{t}\left(t\right)*h_{\mathrm{sys}}\left(t\right)*h_{\mathrm{fs}}\left(t-\tau_{\mathrm{III}}\right)*\left(h_{1}\left(t\right)+h_{3}\left(t\right)\right)$$

As the unique coding filter transfer functions superimpose equally, this signal is comparable to scenario II. Therefore, a simple spectral analysis cannot discriminate between scenario II and scenario III. Unfortunately, this may lead to ambiguities. For further discrimination, both scenarios’ features, e.g., the time course of the radar measurement, must be taken into consideration. Moreover, in real scenarios, the target RCSs will not be equal and it is very unlikely that both signals are weighted with the same antenna gain. This will result in different behavior, making the target discrimination and classification easier.

#### 2.2.4. Scenario IV

This scenario also provides two targets within the same range resolution cell. However, the cross-range distance between target #5 and target #6 is closer as target #5 is located within the overlap area of the sub-channels #1 and #2, while target #6 is located in the main beam area of sub-channel #2. Comparably to scenario III, the time gating operation does distinguish between both targets so that a coding filter superposition is observed once more:(10)$$s_{r,\mathrm{IV}}\left(t\right)=s_{t}\left(t\right)*h_{\mathrm{sys}}\left(t\right)*h_{\mathrm{fs}}\left(t-\tau_{\mathrm{IV}}\right)*\left(h_{1}\left(t\right)+2\cdot h_{2}\left(t\right)\right)$$

In this scenario, the spectral analysis of $s_{r,\mathrm{IV}}\left(t\right)$ will deliver features from both coding filter transfer functions $H_{1}$ and $H_{2}$. However, as target #5 provides both signatures and target #6 solely the signature of $H_{2}$, the latter characteristic is more distinct. This may lead to the interpretation that the observed range cell provides two targets, which are located in sub-channel #2 ’s main beam and within the overlap area to sub-channel #1.

#### 2.2.5. Scenario V

Scenario V contains two targets within the same range resolution cell. Consequently, the separation of target #7 and target #8 by using the aforementioned time gating operation is not possible. Therefore, the coding filter functions superimpose, leading to the following RX signal:(11)$$s_{r,V}\left(t\right)=s_{t}\left(t\right)*h_{\mathrm{sys}}\left(t\right)*h_{\mathrm{fs}}\left(t-\tau_{V}\right)*\left(h_{1}\left(t\right)+h_{3}\left(t\right)\right)$$

The spectral analysis of $s_{r,V}\left(t\right)$ will reveal both coding filter transfer functions of filter #1 and #3. However, due to the known antenna arrangement, no intersection area is present so the superposition of two coding functions of non-neighboring areas leads to the interpretation of two separate signals in the corresponding regions.

## 3. Signal Processing

As mentioned before, the measured signal is a 1-dimensional signal, which does not directly provide information on the investigated two-dimensional scene. In order to assign the detected targets to their cross-range direction, the unique target features must be extracted and classified as shown in Figure 3. Here the measured signal $s_{\mathrm{IF}}\left(t\right)$ is Fourier transformed (FT) to detect the targets in the radar diagram. After that, feature extraction is performed via time-gating of the spectrum of $S_{\mathrm{IF}}\left(f\right)$.

Within the time-gating block the detected targets are first separated using a peak finder algorithm. The location of the detected targets is represented by a delta pulse, which is then convolved with a window function. The convolution thus leads to a shift of the window to the location of the target. The targets are then cut out of the spectrum by multiplying $S_{\mathrm{IF}}\left(f\right)$ with the window function $H(f-f_{p})$. The time gated target is then transformed with an inverse Fourier transform (IFT) into the time domain, whose time sequence contains the transfer function of the filters. The time sequence of |x(t)| thus serves as the basis for the classification of the cross-range direction. Due to the duality between time and frequency in the used FMCW radar, the frequency components can be mapped to the temporal sample points of the extracted transfer functions, which helps to identify the resonant frequency components of the filter. Assigning the extracted transfer functions to the class $\Omega_{i}$, either unsupervised or supervised methods can be used, which are described in more detail below.

### 3.1. Unsupervised Classification

In this section, the unsupervised classification method is presented. We will focus on a simplified separation in a cross-range direction by reducing the scenario to a single-target environment with only two antennas (a1 and a2) and two filters (8.5 GHz and 11.5 GHz). The simplification thus leads to a distinction between the following cases:(1).Location of the target in the main beam of antenna a1 (notch at 8.5 GHz)(2).Location of the target in the main beam of antenna a2 (notch at 11.5 GHz)(3).Location of the target in the overlap of antenna a1 and a2 (notches at 8.5 GHz and 11.5 GHz).

In order to analyze the corresponding features of interest, the feature space is spanned over the frequency components in the region of the resonant frequencies (see Figure 4a–c).

The absolute values of the power levels of the features A and B are then converted into a probability of occurrence of the respective frequency points. Thus, a notch in the feature space is determined by the distribution of its cumulative quantity. As can be seen in Figure 4d–f, the distribution of the data in the feature space is a direct measure of the probability of occurrence of a notch. In a *n*-dimensional feature space the respective features can thus be determined as a random variable $\vec{x_{n}}$ by the random vector X.
(12)$$X=\left(\begin{array}{c} \vec{x_{1}}\\ \vec{x_{2}}\\ \vdots \\ \vec{x_{n}} \end{array}\right)$$

Since the distribution of X is given by the covariance matrix, the notches can be analyzed via
(13)$$\Sigma_{X}=\left(\begin{array}{cccc} \sigma_{1}^{2} & \sigma_{12} & \cdots & \sigma_{1n}\\ \sigma_{21} & \ddots & & \vdots \\ \vdots & \; & \ddots & \vdots \\ \sigma_{n1} & \cdots & \cdots & \sigma_{n}^{2} \end{array}\right).$$

Here $\sigma^{2}$ indicates the variance along the *n*-dimensional feature space. To investigate the correlation of the features $\vec{x_{n}}$, the eigenvalues of the covariance matrix $\Sigma_{X}$ have to be taken into account. The eigenvalues λ are calculated by
(14)$$\det(\Sigma_{X}-I\lambda )=0,$$

This leads to a characteristic polynomial, whose zeros thus lead to the desired eigenvalues λ. A low eigenvalue thus indicates the occurrence of a notch. If a uniform distribution of $\Sigma_{X}$ between two features occurs, both features contain a notch [21,22].

### 3.2. Supervised Classification

For the supervised classification method a so-called long short term memory (LSTM) network was used, which can be found in Figure 5.

The network was chosen because it is not only suitable for the extraction of stationary features, but also considers the temporal dynamics of tracked objects. As can be seen, each LSTM layer is divided into three different gates: the forget, input, and output gates. The forget gate has the function to erase irrelevant information from the input data. Since LSTM networks are a special type of recurrent neural network, the input data of the network consists of the actual input signal $x_{k}$ and the previous cell states $h_{k}-1$. The sigmoid function σ is then used in order to decide which information is to be forgotten. Forgetting the irrelevant information takes place via multiplication with the cell state $C_{k}-1$ of the previous LSTM layer. In contrast to the forget gate, the input gate subsequently determines which part of the input data should be used to modify the memory of the cell state. The update of the new cell state is then obtained via the addition of the forget gate and the input gate. Finally, the output gate passes the updated cell state $C_{k}$ and the output state $h_{k}$ to the next layer. The algorithm used to train this network was the stochastical gradient descent with momentum (SGDM) method, which can be formulated as follows [25]
(15)$$\hat{w}\left(n+1\right)=\hat{w}\left(n\right)-\eta_{1}\nabla \varepsilon \left(\hat{w}\left(n\right)\right)+\gamma \left(\hat{w}\left(n\right)-\hat{w}\left(n-1\right)\right),\;0\leq \gamma \leq 1.$$

Here $\hat{w}$ represents the parameter vector which is updated with the gradient of the error function ε. Moreover, $\eta_{1}$ is the learning rate and γ determines the influence of how much the previous gradient step is considered in the current iteration. In this method, so-called mini-batches are used in order to train the network. These mini-batches are simply a subset of the entire training data. The use of mini-batches leads to noisy estimations of the optimal parameter vector changes, which determines the training updates to be stochastic. Since the error function ε has to be minimized, the noisy estimation of the parameter vector offers the advantage that local minima can be avoided more efficiently in order to reach a global minimum. Furthermore, the momentum term γ reduces oscillations along the path of steepest descent towards the optimum and favors an acceleration of the training, allowing a faster convergence.

Even though the features are extracted within a black box, they can be classified by a higher level of abstraction, depending on the number of layers. In addition, compared to the unsupervised method, the entire frequency spectrum of x(t) is considered and not only the feature space in which the notches are located.

## 4. Components and System

### 4.1. Coding Filters

#### 4.1.1. Filter Selection

The coding filters are used to imprint the signature of the respective receiving path into the radar signal. By evaluating the transfer function, the received signal can then be correlated to a specific path. In this process, most of the received energy must be preserved to not affect the range information of the target reflection. Therefore, narrow-band notch filters are to be used as filters. For the bandwidth of the notch, a compromise has to be found between the energy loss of the received signal and the detectability of the coding. Another aspect is the number of coding channels to be used and the available radar bandwidth.

Since a bi-static radar system will be used, the matching of the filters is not critical. However, reflectionless filter topologies are advantageous to suppress interfering reflections in the receive path and coupling between neighboring antennas. As shown in [26] there exists a type of filter that provides both perfect match and nearly infinite stop-band attenuation by canceling the waves of two signal paths by superposition. Later, an even simpler design with coupled resonators was introduced in [27]. By scaling the line lengths, the filter type can be applied to X-band frequencies.

#### 4.1.2. Simulation and Measurement

To perform the radar measurements, two notch filters with stop-band frequencies of about 8.45 GHz and 11.33 GHz were designed. Figure 6 shows the simulation model as well as the fabricated filter structure of the 8.5 GHz filter. Rogers RO4350B with a thickness of 0.254 mm and tanδ=0.0037 was used as substrate. It should be mentioned that the dielectric losses are 4 times later than in the substrate used in [27]. As it turned out, additional losses due to surface roughness were assumed to be too low in the initial filter design. This resulted in a smaller stopband attenuation of the realized filters. The simulation model was later adjusted to match the observed filter characteristic. In further optimization steps, a stop-band attenuation of up to 20 dB could be achieved. Simulated and measured S-parameters of the realized filters are shown in Figure 7.

The overall shape of the transmission factor agrees well with the simulation. The losses outside the stop-band are moderate with 0.7 dB and are to be expected due to the frequencies involved and the choice of substrate. The stop-bands are located at 8.49 GHz with 7.4 dB attenuation and 11.44 GHz with 7.1 dB attenuation. The response is very smooth and the notch is symmetrical. Both filters show a small shift of the resonance towards higher frequencies. This can be explained by manufacturing tolerances. Under etching in the magnitude of the copper thickness, i.e., tens of micrometers, is rarely avoidable and is already sufficient to shift the resonant frequency in the observed scale. This offset could be taken into account in the simulation model and thus be corrected, as long as the undercut remains constant in production. The line widths are also affected by these tolerances, and so are the gap widths. These parameters have an influence on the matching of the filter structure. Simulation and measurement therefore deviate slightly from each other. However, the average matching is still achieved and is very good with $S_{11}<-20$ dB for the 8.49 GHz filter and $S_{11}<-25$ dB for the 11.44 GHz filter.

### 4.2. Antennas

#### 4.2.1. Antenna Selection

The choice of the antenna type determines decisive properties for the presented radar system, such as the mechanical size of the antenna array, the scalability to N channels as well as the achievable resolution in range and cross-range direction. In order to clearly define the sectors and the overlap areas, the antenna should have a strong main lobe and side or back lobes that are as well suppressed as possible. The width of the main lobe depends on the application and the desired cross-range resolution or the number of antennas. The half-power beamwidth (HPBW) should be set so that targets in the overlap area are still sufficiently detected. However, the side lobes should already be clearly suppressed in the main lobe of the neighboring antenna in order to clearly distinguish the radar scenarios shown. In the later shown measurements for cross-range detection, an angle of up to 90° should be detected. Therefore the antennas are to be aligned at an angle of 60° to each other. In order to still sufficiently illuminate the overlap area between the antennas, the HPBW in the cross-range direction is also to be selected at 60°.

The evaluation requires time gating of the received signals. To ensure that the time window is sufficiently small and targets can be separated from each other in range direction, the corresponding radar bandwidth must be ensured. In combination with an overall simple system design, this usually requires a high relative bandwidth, which cannot be realized with all antenna types. On the one hand, the antenna should exhibit sufficient matching within the bandwidth used in order to minimize reflections within the system. On the other hand, a reasonably flat gain is desirable so that constant system behavior can be assumed for the coding in the frequency domain. The radar system used to perform the later measurements operates in the frequency range of the X-band from 8.12 GHz. Thus, the absolute bandwidth is 4 GHz and the relative bandwidth is 40 %.

The most common antenna shape for ultra-wideband radar systems of highest performance is the horn antenna. It belongs to the group of aperture antennas and represents a modified open waveguide radiator. The waveguide end is extended with a horn structure, which gradually matches the wave impedance of the waveguide to the free space impedance. On the one hand, this reduces reflections at the end of the antenna. On the other hand, it increases the radiating aperture and thus the gain of the antenna. The pitch of the horn structure and the size of the aperture can be used to adjust the directivity and find a compromise between performance and mechanical size. The losses in the antenna are very low because air is used as dielectric and the metallic walls can be realized with good conductivity and high surface quality. Due to the well-defined aperture area, horn antennas usually have very low back lobes, which means high isolation between adjacent antennas. The horn antenna is usually fed via a waveguide adapter, so simple coaxial cables can be used for the connection. While waveguides and couplers are widely available, it may be difficult to obtain horn antennas with suitable parameters. Special fabrication is possible, but this increases the already high costs due to the necessary mechanical precision. Especially for the application in the presented antenna array, these costs are an important aspect of antenna selection. Nevertheless, the main disadvantage of horn antennas is their mechanical size and weight.

An alternative is planar antenna structures, which can be realized on printed circuit boards and thus can be manufactured in common PCB processes. For RF applications special substrates are used which have a defined permittivity and low losses up to the 10 GHz range. These substrates are expensive compared to FR4, but the cost is still significantly lower than the mechanical fabrication of a horn antenna. In addition, a large number of antennas can be fabricated simultaneously, which significantly reduces the cost of a single antenna. The feeding of a planar antenna can be done in many ways. Usually, coaxial connectors are used, which couple directly into a microstrip line. Since the antenna can be integrated on the same substrate as the RF circuitry, this method is very common, especially in mass production. One example is patch antennas in automotive radar systems. Due to the high frequencies of 24 GHz or 77 GHz, large antenna arrays can be integrated into a small area. These achieve a high gain and the radiation pattern can also be adjusted via the array configuration. The high frequency range allows a comparatively large absolute bandwidth despite the relatively narrow-band patch antenna. They are therefore less suitable for use in radar systems with a large relative bandwidth.

Instead, so-called Vivaldi antennas or tapered slot antennas have found widespread use in ultra-wideband applications. It was first introduced in 1979 by P.J. Gibson and consists of an exponentially expanded slot line [28]. The electric field is initially concentrated in the narrow region in the slot and then slowly coupled to the free space by the flare and finally detached. Due to the absence of a resonant structure, the Vivaldi antenna is in principle very broadband. The upper and lower cutoff frequency is determined by the dimension along the slot. The gain can be adjusted via the opening rate and the length. The simple implementation and generally good performance make the Vivaldi antenna ideal for use in the radar system presented.

#### 4.2.2. Simulation and Measurement

In this section, a Vivaldi antenna is modeled in CST Microwave Studio and investigated for its matching and radiation pattern. Again, Rogers RO4350B with $\varepsilon_{r}=3.66$ and a thickness of 0.254 mm is used as substrate. Figure 8 shows the basic antenna structure. On the bottom side is the ground plane with the tapered slotline. The antenna is fed via the SMA end launch connector on the left side. This feeds into a microstrip line with an initial impedance of 50 Ω. The microstrip line is routed across the gap and terminated with a λ/4 open stub, creating a current maximum in the gap area and efficiently coupling the energy into the slot. The slotline is similarly terminated on the left side with a λ/4 shorted stub. This causes the energy to propagate in the direction of the taper. The circular implementation of the stub allows for a wider bandwidth match. To minimize the impedance change at the transition to the slotline, the linewidth and thus the impedance of the feed line is reduced evenly until both impedances are equal. Due to the symmetrical design, a horizontally and vertically symmetrical antenna pattern results.

Figure 9 shows the simulated matching of the presented antenna with the reference plane on the microstrip line. Also shown is the measurement of the realized antenna with the reference plane at the SMA connector. The antenna is well matched between 8.5 and 12 GHz with $S_{11}<-17$ dB. At 8 GHz the matching is reduced to −13 dB. Simulation and measurement agree within the expected range. The additional minima indicate reflections, as they can be caused by the SMA-to-microstrip transition, which was not included in the simulation.

From the distribution of the E-field in Figure 10, the expansion of the field components along the tapered slot can be observed. In addition, field components are visible, which propagate around the ground plane and radiate sideways and backward. This is part of the reason for the limited suppression of the side lobes. These field components can be further attenuated with corrugated structures [29] or absorber material at the side edges.

The fabricated antenna was fully characterized by an antenna measurement system. Figure 11 shows the absolute gain determined in the process, including losses and mismatch. There is a notable variation with frequency, but this is in rough agreement with the simulation. The gain increases with rising frequency, as the antenna size-to-wavelength ratio increases accordingly.

Figure 12 compares the antenna pattern in the E-plane. Measurements were taken at the center frequency of 10 GHz in the main direction, so θ=ϕ=0. The shape of the main lobe agrees very well with the simulation. The half-width is about 68°. The position of the side lobes varies, but the average magnitude is in the range of the simulated values. Small deviations due to the measurement setup are possible as well.

Due to the very broadband radar measurements, the frequency dependence of the antenna parameters must be taken into account. In general, the width of the main lobe of a Vivaldi antenna decreases with rising frequency and the gain increases accordingly, as can already be seen in Figure 11. The size and position of the side lobes also vary. Due to the linear frequency sweep of the FMCW method and the subsequent signal processing, all received signals are integrated. This represents an averaging of the antenna parameters over frequency. Figure 13 shows the influence of such an averaging on the gain. Side lobes and zeros balance each other out, resulting in a very flat response outside the main lobe and a side lobe suppression of about 10 dB. The gain in the main direction corresponds directly to the average value of the gain from Figure 11 measuring 6.6 dBi.

In summary, a small and cost-effective solution was found with the realized Vivaldi antenna. The antenna shape can be arbitrarily adapted to the requirements of the specific measurement scenario. Manufacturing in common PCB processes is possible, which means that a larger number of antennas can also be used for complex arrays without any problems. The performed measurements agree very well with the preceding simulations and the antenna shows good performance for broadband applications.

## 5. Verification Measurements

### 5.1. Measurements Scenario and Setup

The results of the following outdoor measurement campaign investigate the performance of the proposed, novel cross-range enhancing radar method. The measurements have been performed at the ballpark of the local baseball club. As a radar device, we utilized an X-band FMCW Radar system with separate transmit and receive paths. Because of its perfect 3 dB beamwidth of 90 degrees, we choose a WR-90, open-ended, rectangular waveguide as the TX antenna. In order to cover the whole AOI, the main beam direction was pointed to the field’s center direction, i.e., 2nd base. Moreover, two of the above-mentioned Vivaldi antennas were used as RX antennas. Together, they cover the 90-degree azimuth angle of the field by subdividing it into three beam areas namely: Center-Beam 1 (CB1) and Center-Beam 2 (CB2) for the fringe areas providing each a single coding filter function. Their overlap area, which results in the coding of both filters, is denoted as Overlap-Beam (OB) further on. Within the infield, five target positions have been identified that are located in the different beams as well as in different LOS-range distances. Figure 14 shows drone footage of the test field with overlaid measurement sectors, indicated target positions, and corresponding coordinates. Here, the origin of the coordinates was set to the radar device at the home base.

Furthermore, Table 1 lists all targets T1–T5 and their properties including corresponding field coordinates, LOS distance to the radar device, and their sector beam locations.

### 5.2. Measurement Results

In order to test and verify the above-described operation theory, five different scenarios have been set up and were investigated within the measurement campaign. The different scenarios are listed in Table 2, while the entries “1”and “0” represent the presence or absence of the corresponding targets T1–T5, respectively. For checking the field coverage and the setup’s time gating operability, all targets were put into the field. As we did not perform any target classification in this scenario, it is labeled as scenario #0.

Figure 15 shows the measured radar diagram of the scenario #0. Due to the length of the cables, all targets are located 80 cm further away than their corresponding field coordinates but can be clearly observed at their designated positions. Next to the range distribution, it is possible to investigate the targets’ spectra by time gating the targets (see Figure 16) and feed the corresponding data to the inverse Fourier transformation.

Figure 17a–c show the resulting spectra of the targets T1, T3, and T4, respectively. As can be seen, the notch at 11.5 GHz can be clearly detected for T1. The same is valid for the two notches at 8.5 GHz and 11.5 GHz for T4, which has both features due to its position in the overlap region. Unfortunately, no clear notch can be detected for the target T3, because the received power of the target is too low and too much affected by ground clutter. However, these results show that the concept works in principle since the notches in the main beam direction as well as in the overlapping area of both antennas can be extracted.

### 5.3. Detection Accuracy

Using the measured data and the given a priori information of the different measurement scenarios, the accuracies indicating the probability of the cross-range direction of the target were calculated. For this purpose, the neural network was trained first, and its classification accuracy was then compared with the accuracy of the unsupervised classification method. For the training of the neural network 1188 data sets were used, which were measured in a predefined test environment. Furthermore, 297 datasets of the measurement scenario #0 were used for the validation of the training. The comparison of the training process and the corresponding classification accuracy of the validation data is shown in Figure 18.

As can be seen, the training process reaches 100 percent accuracy after approximately 85 iterations and achieves a very good classification probability of the cross-range direction with 100% validation accuracy. Based on these results, the fundamental proof of the suitability of the concept is provided. With the help of additional training data in different measurement scenarios, the precision of the method could be further improved at this point. The subsequent classification results of the unsupervised and supervised methods for scenarios I–IV can be found in Table 3. For the calculation of the detection accuracies, the classified directions were used and compared to the true direction. Based on the correctly classified directions the mean value (in %) was determined. The training and the calculation of the classes was carried out with the help of MATLAB using an *Intel(R) Core(TM) i7-9700 CPU*. With this setup the training required 3 minutes and the calculation time of the neural network classification required 0.6 seconds. The calculation of the unsupervised method required 0.9 seconds and was therefore slightly slower than the supervised method. However, we expect that the unsupervised method will slow down for an increased number of directions, because of its unparallelizability structure.

Table 3 shows that the supervised classification has higher accuracy than the unsupervised classification. This is due to the fact that the extraction of the features is more robust towards calibration errors of the radar system. These calibration errors are already taken into account during the training of the network since the training data also contains these errors.

## 6. Discussion

The presented results of the measurement campaign reveal the applicability of the proposed radar concept. The introduced and utilized measurement equipment was capable to detect and measure all targets within the described scenarios. Furthermore, it was shown that all time-gated targets carried the corresponding notch filter transfer functions of their sub-channel, leading to a successful direction alignment of every single target. As this observation is valid for center beam and overlap beam targets, the enhancement of the cross-range resolution was proved as well. Moreover, it was further observed that recorded clutter lead to an unpredicted frequency behavior. Regarding the presented signal processing methods, both single target scenarios were classified with high accuracy of 85.5% and 100% for the center beam and overlap beam scenario, respectively. In the future, these results can be further improved by considering additional training data. In contrast, the unsupervised method already revealed lower classification accuracies with only 38.89% and 22.73% for the mentioned scenarios. The lower accuracies are the result of the already mentioned additional clutter, which caused interference effects in the frequency domain leading to confusion with the notches of the intended frequency coding. Here, different coding filters with improved coding design abilities might improve the classification accuracy of the unsupervised method. In terms of the multi-target scenarios, a further classification accuracy decrease to 60.6% was observed for the supervised method when the responses of two targets located in a center and in an overlap beam are superimposed. This is the result of further interference effects but can be tackled by considering additional training data in the future. As predicted in the theory section, the overlap of two identical targets in identical LOS distances located in two main beam areas lead to the confusion of the result classes so this scenario was not classified at all. However, depending on the radar application this worst-case scenario will not appear very often or for a long time when targets move independently. In order to overcome the drawback anyway, one might consider modulating the beam distributions by activating and de-activating corresponding sub-channels. The unsupervised classification method failed for both multi-target scenarios, which clearly reveals the advantage of the supervised method. In future work, the classification methods must be optimized by considering further result classes and more divergent training data. Furthermore, the coding filter design can be improved for avoiding confusion with interference effects. Finally, calibration methods should be implemented for reducing the system’s frequency behavior.

## 7. Conclusions

In this contribution, we introduced a novel radar method for enhancing the cross-range resolution. The proposed method makes use of frequency-coded sub-channels that enable the scanning of the whole area of interest with a single radar measurement while preserving the maximum range resolution. This proposed operation theory leads to a number of different single and multi-target scenarios that propose different classification accuracies. For investigating the basic concept, different components, i.e., Vivaldi antennas and reflectionless notch filters, have been introduced and their performances were shown in simulations and measurements. Furthermore, we presented two different signal processing schemes for target classification which utilize supervised and unsupervised methods. Finally, results of an open field measurement campaign were shown, which proved the applicability of the novel radar concept and presented hardware. Moreover, it was shown that the proposed signal processing for target classification already delivers good results.

## Figures and Tables

**Figure 1 sensors-22-03343-f001:**
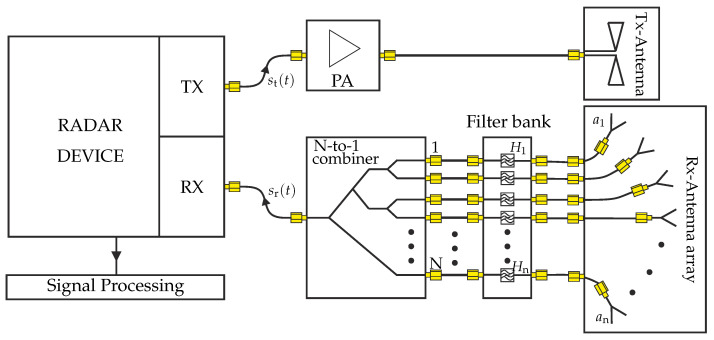
Block diagram of the proposed bi-static radar setup with RX-channel coding.

**Figure 2 sensors-22-03343-f002:**
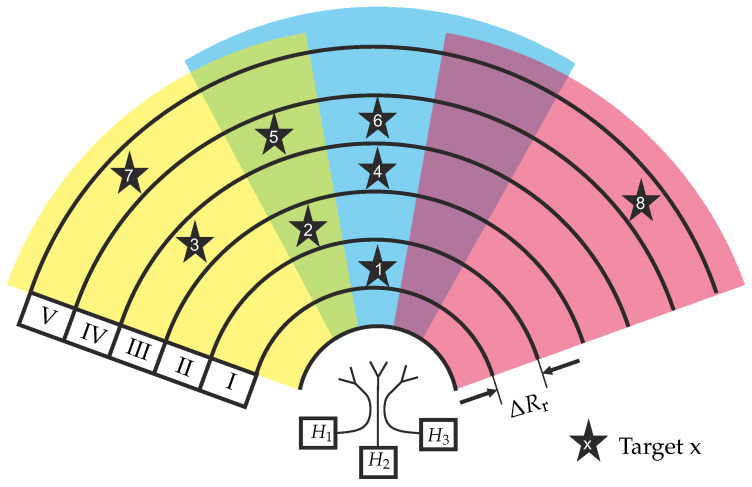
Illustration of different radar scenarios occurring for coded RX- channel overlay operation.

**Figure 3 sensors-22-03343-f003:**
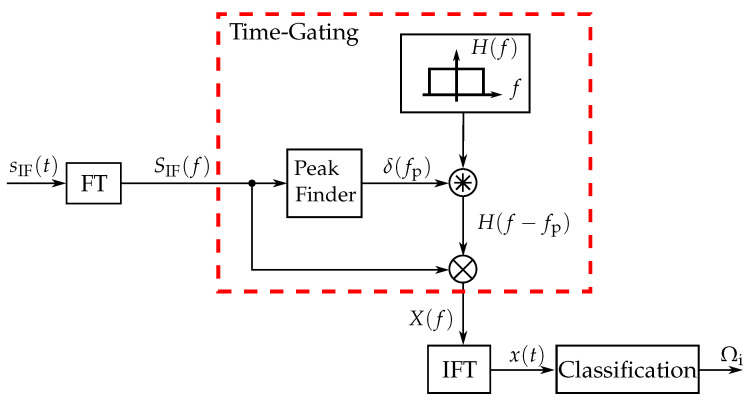
Signal flow diagram of the feature extraction and classification [20].

**Figure 4 sensors-22-03343-f004:**
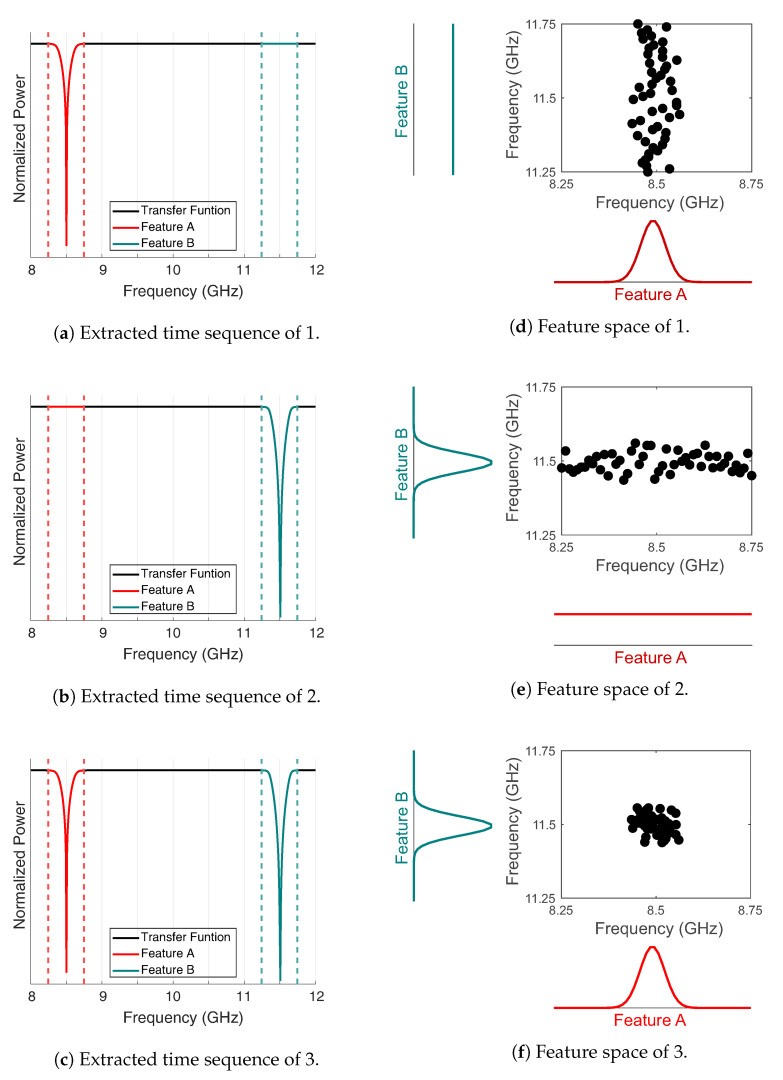
(**a**–**c**) Extracted time sequence and (**d**–**f**) associated feature space.

**Figure 5 sensors-22-03343-f005:**
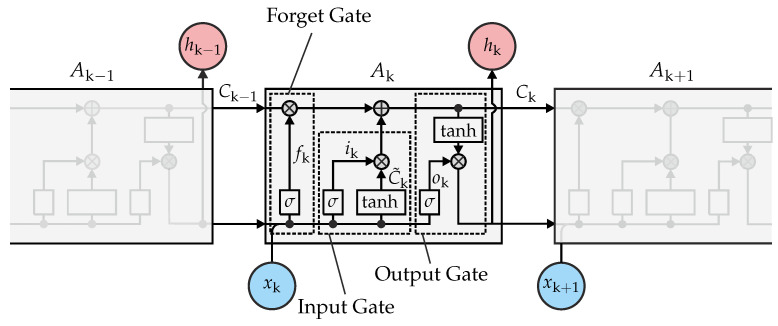
Block diagram of an LSTM network [23,24].

**Figure 6 sensors-22-03343-f006:**
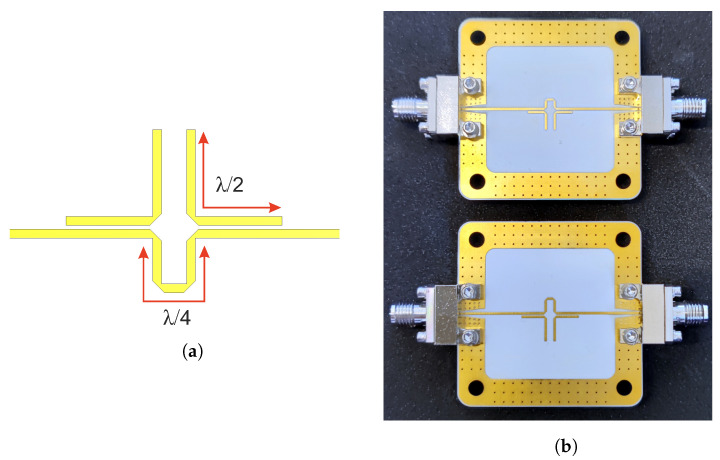
(**a**) Simulation model and (**b**) manufactured notch filters with end launch SMA connectors.

**Figure 7 sensors-22-03343-f007:**
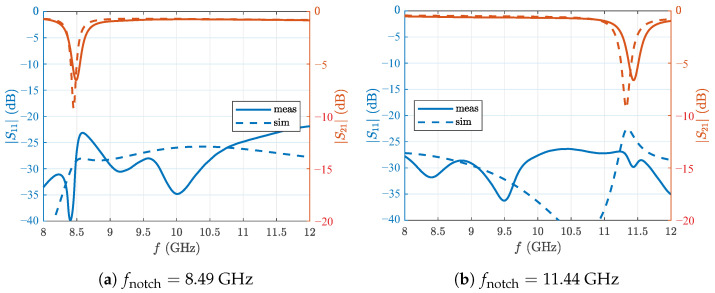
Simulated and measured S-parameters of the proposed notch hlfilter.

**Figure 8 sensors-22-03343-f008:**
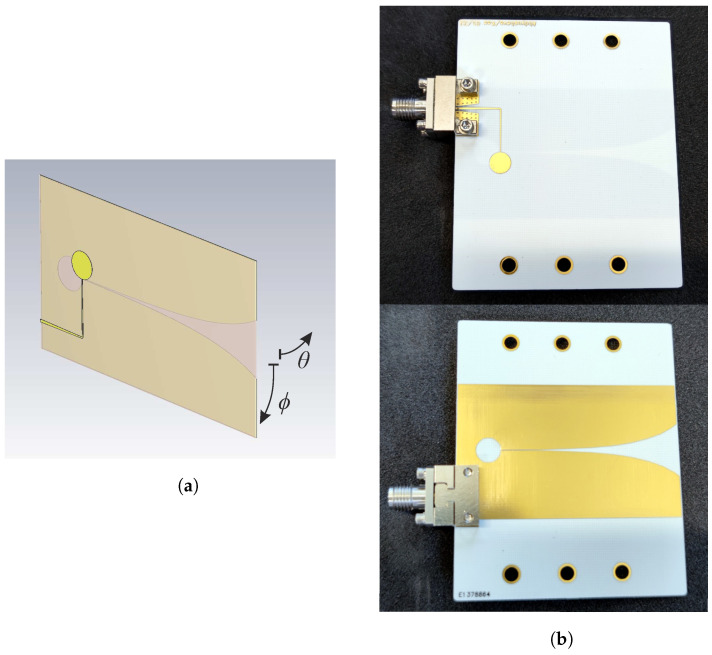
Simulation model (**a**) and manufactured Vivaldi Antenna (**b**) with end launch SMA connectors.

**Figure 9 sensors-22-03343-f009:**
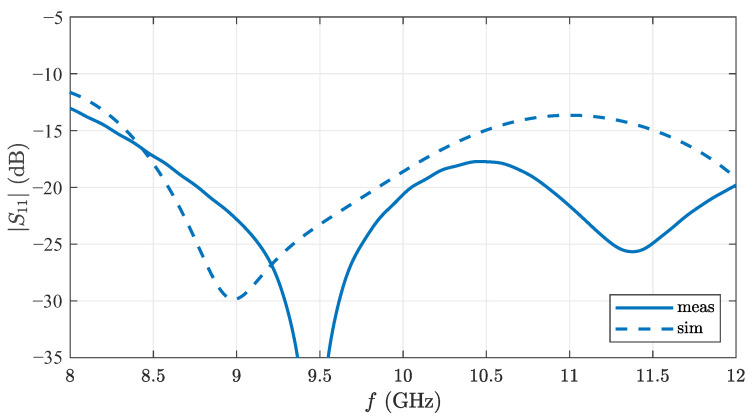
Simulated and measured S-parameters of the proposed Vivaldi antenna.

**Figure 10 sensors-22-03343-f010:**
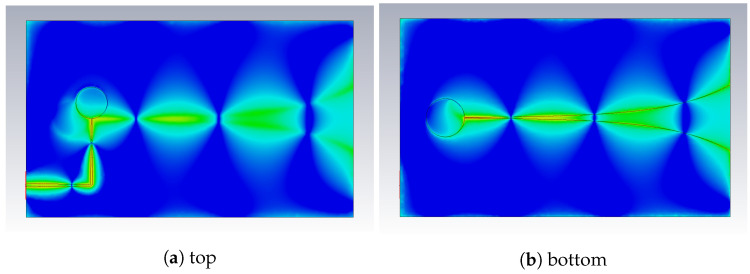
Simulated E-field on the antenna surface.

**Figure 11 sensors-22-03343-f011:**
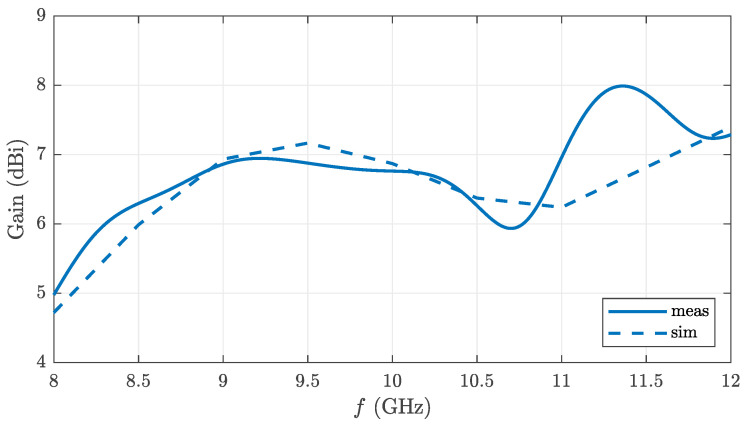
Simulated and measured gain of the main lobe versus frequency.

**Figure 12 sensors-22-03343-f012:**
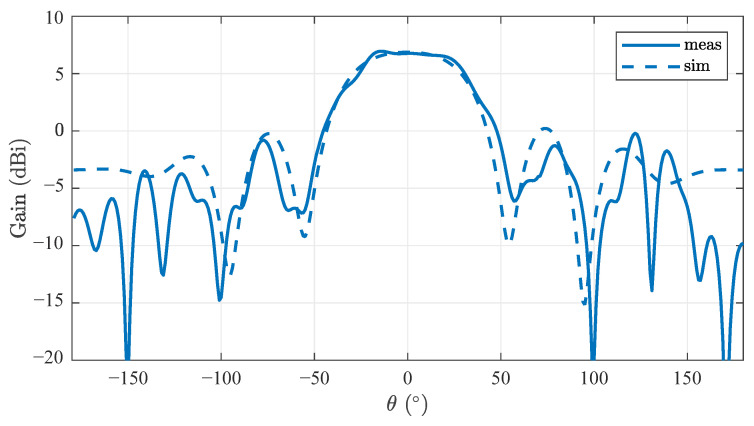
Simulated and measured antenna pattern at center frequency of 10 GHz and Φ=0.

**Figure 13 sensors-22-03343-f013:**
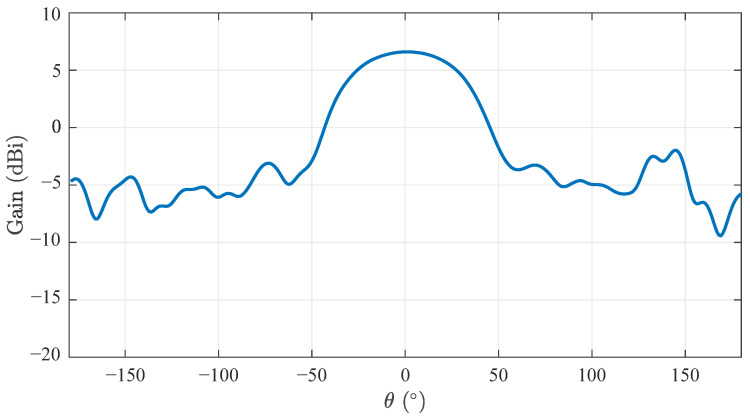
Measured antenna pattern averaged over the frequency range of 8…12 GHz and Φ=0.

**Figure 14 sensors-22-03343-f014:**
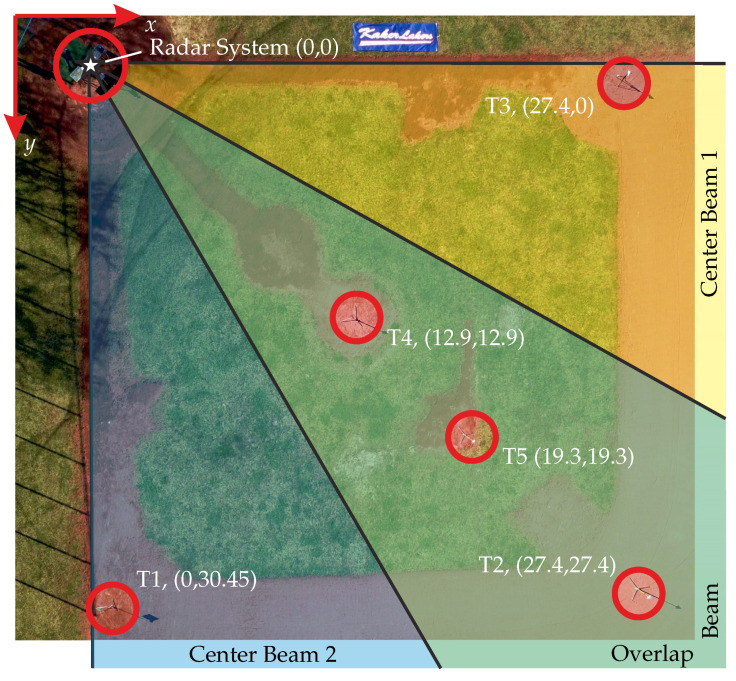
Drone footage of the measurement scenario with overlaid antenna beam areas and indicated target locations. Coordinates (x,y) of the targets T1–T5 are given in meters.

**Figure 15 sensors-22-03343-f015:**
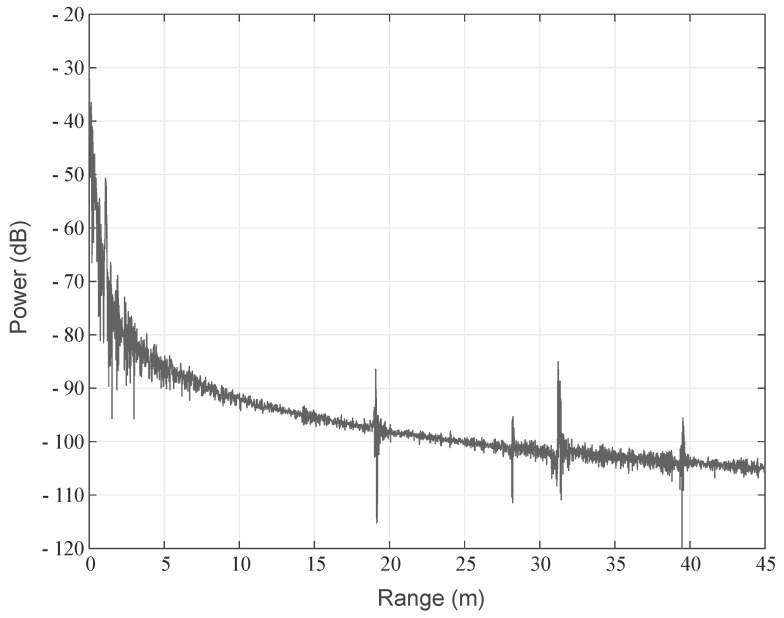
Measured radar diagram of scenario #0.

**Figure 16 sensors-22-03343-f016:**
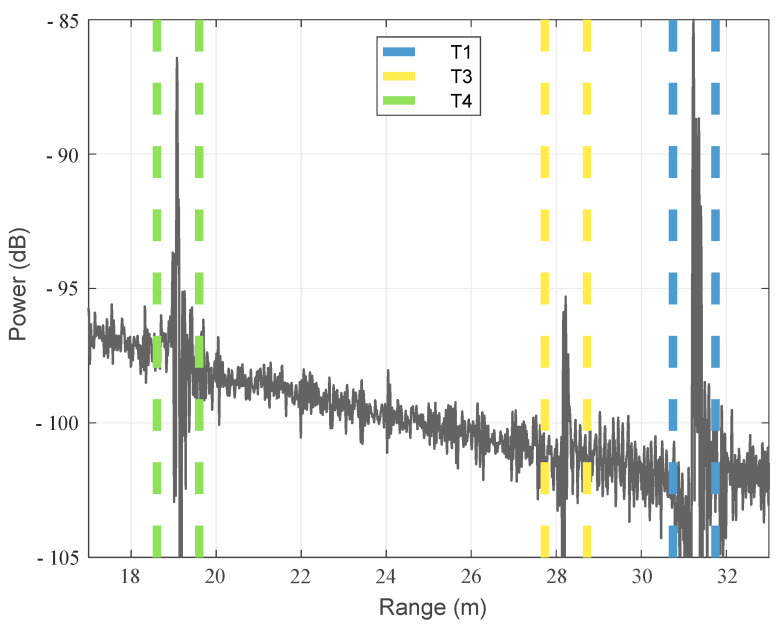
Measured radar diagram of scenario #0 with indicated time gates.

**Figure 17 sensors-22-03343-f017:**
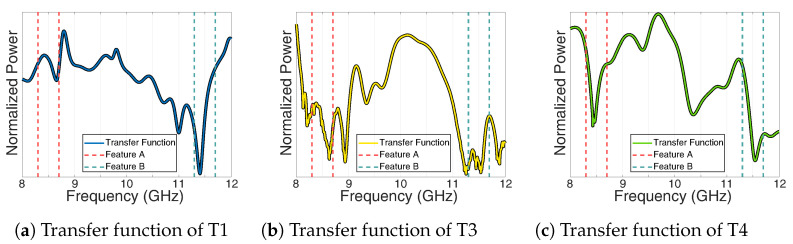
Extracted transfer function of the time gated targets with indicated cross-range features.

**Figure 18 sensors-22-03343-f018:**
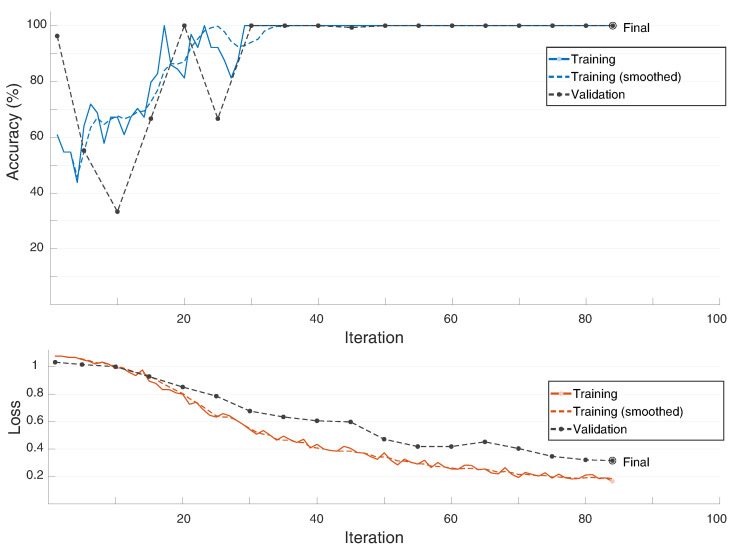
Plot of the LSTM-training progress using validation data of the measured scenario #0.

**Table 1 sensors-22-03343-t001:** Target plan and properties.

Target No.	Coord. (x,y) in m	LOS-Distance *R* in m	Beam Area
T1	(0, 30.45)	30.45	CB2
T2	(27.4, 27.4)	38.75	OB
T3	(27.4, 0)	27.4	CB1
T4	(12.9, 12.9)	19.1	OB
T5	(19.3, 19.3)	27.4	OB

**Table 2 sensors-22-03343-t002:** Measurement scenarios.

Scenario No.	T1	T2	T3	T4	T5
0	1	1	1	1	1
1	1	0	0	0	0
2	0	0	0	1	0
3	1	0	1	0	0
4	0	0	1	0	1

**Table 3 sensors-22-03343-t003:** Table of detection accuracies.

Scenario No.	Accuracy (Unsupervised Method)	Accuracy (Supervised Method)
I	38.89%	85.5%
II	22.73%	100%
III	0%	0%
IV	0%	60.6%

## Data Availability

Not applicable.

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
