# Peer review of "Enhancing the Radar Cross-Range Resolution in Ultra-Fast Radar Scans by Utilizing Frequency Coded Sub-Channels"

_sensors, 2022, doi:10.3390/s22093343_

Round 1

Reviewer 1 Report

This paper handles with a single-channel radar method when use the frequency coded subchannels for enabling better cross-range resolution. The whole area of interest is scanned within a single radar measurement. To enhance the resolution, the sub-channels’ antenna beams are overlaid, resulting in multiple coding signatures. The authors used several radar scenarios justifying high detection properties and possible applications of their method.

Comments:

  1. This reviewer thinks that current manuscript has some intersections with previously presented conference paper by the same authors. The authors referred their conference paper as in-print  paper. The authors should explain the principal theoretical contributions of this manuscript paper and compare it with previously presented conference paper and justify the intersections of these two papers.
  2. In opinion of this reviewer, the authors should present much more explications for the used supervised classification method that is based on LSTM network. The authors referred papers 22, 23 but for better understanding by potential reader they should explain the details of the used network.
  3. Some small grammar and stylistic errors in English language should be corrected with help of native speaker.

Author Response

Thank you for your time and effort. Please check the attached file for our answers

Reviewer 2 Report

REVIEW

Article titled:  “Enhancing the radar cross range resolution in ultra-fast radar

scans by utilizing frequency coded sub-channels

Sensors  no. 1673686

List of Authors:

Christoph Baer, Nicholas Karsch, Robin Kaesbach, Thomas Musch

  1. In this article, the Authors present handles a single-channel radar method that utilizes frequency coded sub-channels for enabling cross-range resolution.

In Section no. 1 the fundamentals of the radar channel frequency coding method is discussed. In Section no. 2 the Authors show essential hardware components, i.e., antennas, coding filters, and their performance is demonstrated in simulations and measurements. In the next section two different signal processing methods for target classification are presented. And finally, measurement campaign, that investigates several radar scenarios, is shown by the Authors.

  1. In Introduction, the Authors made extensive literature review dealing with the most important aspects about concept of a dielectric hemispherical lens antenna for polarimetric radar applications, design of a high gain antenna system with wide range beam steering capability for broadband radar applications, characterization of a beam steering lens antenna for industrial radar measurements, monopulse antenna in SIW technology for beam steering applications, electro-mechanically tunable meta-surfaces for beam-steered antennas, target localization based on high resolution mode of MIMO radar, SAR imaging using MIMO frequency modulated continuous wave sensors, 3D imaging radar based on FMCW, and many others.
  2. In the process of radar cross range resolution in ultra-fast radar the very important is the process of phase unwrapping. For this reason, optimizing the minimum cost flow algorithm for the phase unwrapping process, and adaptive forming the beam pattern of antenna is very important. In my opinion, the Authors should read the work entitled Adaptive Forming the Beam Pattern of Microstrip Antenna with the Use of an Artificial Neural Network”, Int. J. of Antennas and Propagation, Hindawi, vol. 2012, and include the article in the Reference list.

  1. In Section no. 1 entitled “Concept” the structure of proposed bi-static radar setup with RX-channel coding is presented. Also, single and multi-target scenario for proposed antenna arrangement is shown on the Figure 2. The Figure no 3 shows the diagram of the feature extraction and classification, and the next in Section no 2.2 classification method called long-short-term-memory (LSTM). The Authors used the algorithm network with stochastical gradient descent with momentum (SGDM). My questions are as follows:
    1. Why this kind of algorithm was applied?
    2. The process of classification and extraction of features is not precisely described. The description should be made more detailed.
    3. The simulation was performed in the band frequency 8,5 -11,5 GHz. And how “a different frequency range” will have an impact on simulations?
    4. As the Authors claim, both single target scenarios were classified with high accuracy of 85.5% and 100% for the center beam and overlap beam scenario, respectively. How the classification accuracy was calculated?
    5. What is the computational complexity of these method/algorithm? Computational complexity calculations should be presented in this article.

The article is very interesting, but it requires systematization of the concepts/definition and answers to the questions contained in the review of this article.

The Authors addressed a problem which is relevant and appealing for this journal. However, I cannot recommend the current manuscript for publication unless the current version is corrected. After providing the amendments to the article, the work ought to be reviewed once again.

Author Response

(The authors gave the same response as above.)

Round 2

Reviewer 1 Report

The authors have attended all comments of this reviewer

Reviewer 2 Report

REVIEW_2

Article titled:  “Enhancing the radar cross range resolution in ultra-fast radar

scans by utilizing frequency coded sub-channels

Sensors  no. 1673686

List of Authors:

Christoph Baer, Nicholas Karsch, Robin Kaesbach, Thomas Musch

The article Sensors no. 1673686 entitled “Enhancing the radar cross range resolution in ultra-fast radar scans by utilizing frequency coded sub-channels” has been carefully modified and well revised. The work is supposed to be finally accepted for publication in Sensors.